# Enriching Personalized Endometrial Cancer Research with the Harmonization of Biobanking Standards

**DOI:** 10.3390/cancers11111734

**Published:** 2019-11-05

**Authors:** Meera Adishesh, Dharani K. Hapangama

**Affiliations:** 1Department of Women’s and children’s health, Institute of Translational medicine, University of Liverpool, Liverpool L8 7SS, UK; meeadish@liverpool.ac.uk; 2Liverpool Women’s Hospital, Liverpool L8 7SS, UK

**Keywords:** biobanking, biospecimens, harmonization, translational research, endometrial cancer

## Abstract

Endometrial cancer is the commonest gynecological cancer, with an incidence predicted to escalate by a further 50–100% before 2025, due to the rapid rise in risk factors such as obesity and increased life expectancy. Endometrial cancer associated mortality is also rising, depicting the need for translatable research to improve our understanding of the disease. Rapid translation of scientific discoveries will facilitate the development of new diagnostic, prognostic and therapeutic strategies. Biobanks play a vital role in providing biospecimens with accompanying clinical data for personalized translational research. Wide variation in collection, and pre-analytic variations in processing and storage of bio-specimens result in divergent and irreproducible data from multiple studies that are unsuitable for collation to formulate robust conclusions. Harmonization of biobanking standards is thus vital, in facilitating international multi-center collaborative studies with valuable outcomes to improve personalized treatments. This review will detail the pitfalls in the biobanking of biosamples from women with cancer in general, and describe the recent international harmonization project that developed standardized research tools to overcome these challenges and to enhance endometrial cancer research, which will facilitate future development of personalized novel diagnostic strategies and treatments.

## 1. Introduction

Endometrial cancer (EC) is the 4th most common cancer in women, and it is therefore the commonest gynecological cancer. Around 9000 new cases of EC were diagnosed in the UK in 2013 [1]. Whilst the incidence of many other cancers is reducing, and in general, cancer-associated death rates are declining, the incidences of EC and EC-associated mortality rates are on the rise [2]. In the UK, there has been a 43% increase in age-standardized incidence of EC compared to the 1990s [1], accounting for about 3% of all female deaths (2012). Similarly, UK survival figures indicate the mortality rates from EC have gone up by 21% over the last decade, with a projected rise of 19% by 2035 [3]. The rise in EC rates is a global phenomenon, as shown by European and North American studies due to reasons detailed later in this review. For example, in Norway, the estimated rise in the incidence of EC is 50–100% by year 2025 [4]. Increased efforts into finding new preventative and diagnostic strategies and determining personalized prognostic and therapeutic targets are therefore urgently required in order to reduce the high mortality and morbidity rates associated with EC. 

Our current understanding of the human endometrium is relatively poor, due to its species-specific functional and regulatory differences. For example, regular menstrual shedding, scar-less repair and regeneration, are hallmarks of the human endometrium, but these processes are not seen in most other mammals [5,6]. This precludes the translation of knowledge on endometrial function derived from the other mammals, to the human endometrium. 

Estrogen exerts its trophic/mitotic effects mainly via estrogen receptor alpha (ERα), whilst ligand-activated ERβ counteracts and regulates ERα action [5]. Although endometrium is a target organ for ovarian sex steroid hormones, compared with the other comprehensively researched, hormone responsive organs, there are further striking dissimilarities in the responses of endometrial cells to ovarian hormones in humans vs. other mammals. This is exemplified by the fact that estrogen plays an important trophic role in both breast and endometrial tissue, promoting carcinogenesis. However, tamoxifen, an estrogen receptor modulator, inhibits estrogenic action in breast tissue, and thus is an anti-cancer agent used in breast cancer treatment, while it is an inducer of endometrial growth and EC. This further highlights the urgent need for improving our current understanding of the normal endometrial function and EC, as well as the need for expansion of EC research using human samples. 

## 2. Reasons for the Increase in the Incidence of EC

We aim to highlight the need for harmonizing biobanking in EC in this review. The reasons for the enduring increase in the incidence and the EC-associated mortality are multifactorial [7]. The influence of these factors in a particular biological sample is important and relevant to studies exploring either the pathogenesis or the therapeutic targets of EC. Therefore, whilst such information may not be relevant to other types of cancer, they should accompany EC-biospecimens. Some of these factors are listed below for clarity.

*Obesity* is a significant risk factor for developing EC, and is also responsible for an increase in perioperative morbidity [8]. EC is an *age*-related disease that is commonly present in postmenopausal women [1]. Since the endometrium is exquisitely sensitive to ovarian hormones [5], the *exposure to excessive exogenous or endogenous estrogen* in particular increases the risk of EC [7]. *Hormone replacement therapy (HRT)* that is commonly used to alleviate the menopausal symptoms by peri- and postmenopausal women is associated with an increased risk of EC [9,10]. *Tamoxifen*, a selective estrogen receptor modulator (SERM), is used to reduce the risk of a recurrence of breast cancer. On breast tissue, tamoxifen has anti-estrogenic effects, while moderate estrogenic effects are seen on the endometrium; therefore, in standard doses it causes endometrial proliferation leading to hyperplasia, polyp formation and invasive cancer [11,12,13]. 

Increased, unopposed endogenous estrogen activity in women with *Polycystic ovarian syndrome (PCOS)* increases the incidence of EC by three to four fold, with a lifetime risk of 9% in comparison with 3% in the general population [14]. Hyperandrogenism and peripheral aromatization of androgens, which occurs in adipose tissue and high BMI, are all important features of the PCOS-intensifying estrogenic effect on the endometrium. Late menarche reduces the risk of EC, whilst late menopause increases it [15,16]. In contrast to HRT, the use of hormonal contraception is protective to the endometrium [17]. The reduction of this risk is proportional to duration of use, for every 5 years of use is associated with an RR of 0.76, and this effect persists for about 30 years, and it may be amplified as time progresses [18]. 

*Lynch syndrome* is an inherited syndrome, which is associated with a high risk of colorectal, endometrial, ovarian and urinary tract cancers. Lifetime risk of EC in women with Lynch syndrome is about 60% [19]. EC in these patients when it occurs as sentinel cancers, occurs in younger and low BMI women when compared with sporadic tumors. Patients with medical conditions such as Diabetes, and Parkinson’s disease also have increased predisposition to EC, this may be due to increased insulin resistance or other unknown factors [20].

*Nulliparous* women are at higher risk of EC than multiparous women (nulliparous vs parous: HR, 1.42; 95% CI, 1.26–1.60) [21]. Factors such as increased physical activity and decreased sedentary time are associated with decreased risk of EC [22]. Both former smokers and current smokers have a reduced incidence of EC compared to non-smokers, and this effect can be explained by hormonal modulation affecting hormone-producing organs, adrenals and ovaries [23].

*Increasing life span, obesity* and *a sedentary lifestyle* are global phenomena that will continue to influence the increasing incidence of EC. The presence of all these risks is important to consider when assessing patient samples in EC research. 

They may also influence the clinical success of a study planned for biomarker identification, and thus should be considered initially when collecting bio-samples and also during their analysis. However, unfortunately, many studies have been conducted without these important considerations. This may be the explanation for the frequent observation that many promised biomarkers emerged from initial studies, failing to show sufficient clinical efficiency in larger clinical studies.

## 3. Importance of Patient Derived Samples/Biospecimens in Cancer Research and in Personalized Medicine

Personalized medicine and translational research aim to use clinical and molecular data from individual patients, to develop and validate therapies with greater specificity, thus reducing the number of side effects whist focusing on determining disease predisposition to develop preventative strategies. Human bio-specimens form a crucial link between molecular signatures of an individual’s specific cancer and their response to clinical treatment. Therefore, the information generated from the bio-specimens provides the basis for subsequent treatment [24]. In recent years, the ‘Omics’ revolution has been driving the field of cancer research, providing alternative ways to study biology, heterogeneity and evolution of tumors [25]. Both the genetic background and environmental factors influence the crucial changes in cellular function that result in tumorigenesis, and they also converge to influence the individuals’ risk of developing cancer. Therefore, improving our knowledge in these areas forms the basis of cancer prevention through targeted therapies [26]. For example, epigenetic research depends upon the analysis of biospecimens, and blood and tumor tissue are the commonest types of specimens that have been used. In addition, patient-derived samples, and patient-derived primary cell lines that retain the phenotype and functional characteristics of the parent tumor, are invaluable in a variety of research studies. Functional studies using them may provide more clinically relevant data, such as the response to the chemotherapeutic agents of a tumor, and they will be more representative of the tumor type/population. Therefore, the overall clinical relevance will be high. The internal and external validity of the generated data depends on the use of stringent standards in collecting the biospecimens and the accompanying patient characteristics pertinent to the specific cancer type. This helps researchers to draw direct clinically translatable conclusions, and enables them to tailor the therapeutic options for individualized treatment with the maximum effectiveness, whilst reducing side effects [27]. However, heterogeneity in the collection, processing and storage of the biospecimens can seriously hamper this seemingly straightforward process, leading to questionable molecular integrity of the biospecimens and irreproducible results that impede development of effective diagnostic and therapeutic strategies [24]. 

## 4. Translational, Personalized Research and Role of Biobanks

The main aim of translational research is to accelerate the process of the transition of scientific discoveries from the lab to the patients who will benefit from those findings. Having a sustainable supply of well-documented and high quality biospecimens is a crucial resource for translational research with a specific and personal relevance. Therefore, biobanks form a critical platform, where all such suitable biospecimens are stored for use in research [28]. Disease-specific biobanks have a huge impact on the discovery of bio-markers, therapeutic targets, and in general, for research on treatment of any diseases or specifically of cancers [29]. In this respect, Biobanks are the cornerstone for research, and they are a valuable educational resource, bringing together all the stakeholders in research, and they lead in the validation of standards used even in standard and routine clinical pathology. Biobanks also play a vital role in improving our understanding of epidemiology, pathogenesis and genetics, relevant to particular pathologies. For example, in EC, they provide the means to rapidly embrace the arrival of the next generation of novel technologies in translational medicine, which encompass genomics, proteomics, epi-genomics and metabolomics. 

The main diagnostic approach (e.g., diagnosing cancers and many other diseases) in clinical care has always been the expert pathological scrutiny of the resected tissue and other biological samples routinely collected from patients during their medical procedures. Traditionally, in cancer research, the pathological characteristics identified by histological means are further analyzed using additional techniques such as immunohistochemistry. Therefore, the commonest way the samples are processed and stored is still by preservation in fixative agents such as formalin, followed by paraffin embedding and subsequent storage as blocks. Although this method preserves the tissue architecture for long-time retention, it only allows the consequent use of a limited number of techniques. To examine the functional aspects of a molecule, or to assess the response of a tumor to a chemotherapeutic agent in the laboratory, samples preserved in that way, are not suitable. To rectify this issue, researchers have developed methods suitable for in vitro and in vivo studies that use patient-derived cell lines and freshly collected/freeze-thawed tissue. These can be incorporated into laboratory in vitro models and in-vivo animal models that are preferentially being applied in cancer research. Although these are not the perfect simulation of the in vivo human environment, there is substantial homology, thus they may reduce the need for testing novel therapeutic agents directly in humans and reduce the burden on patients. Therefore, Biobanks, collecting and storing a wide range of different patient specimens (including fresh tissue, fresh frozen tissue, processed tissue, urine, blood or saliva samples and many other specimens), play a vital role in providing valuable patient material for clinically relevant scientific discoveries. Consequently, they support the rapid translation of basic scientific findings to clinical practice for the benefit of cancer patients. 

## 5. Quality of the Biospecimen as a Cause of Bias in Translational Research/Personalized Medicine

A major setback in cancer research at present is the difficulty in identifying clinically effective molecular targets for early detection and for predicting prognosis. Such markers will facilitate efficient stratification of patients for specific treatments, thus personalizing therapy [24]. The reliability of studies investigating this aspect is largely dependent upon the quality and consistency of the standards used for biospecimen handling. The potential variation in collecting, processing and storing different biospecimens, and the accompanying phenotypic and demographic data, [30,31] may lead to different studies providing divergent results that are extremely difficult to evaluate and merge. This lack of uniformity and inadequate adherence to quality standards in biospecimen handling is recognized by the national cancer institute (NCI) as a roadblock in cancer research, thus efforts are being made to overcome this by several international organizations and agencies [32,33,34].

### 5.1. Factors Directly Influencing the Usefulness of a Bio-Specimen

The collection methods, transportation, processing and storage conditions/methods will all determine the final quality of the biospecimen that is being analyzed. Different protocols used in each step of this pathway, from a sample being donated by the patient, to it being received by the researcher in the laboratory, may add a pre-analytic bias to the result obtained from it. Such biases can be introduced prior to the specimens reaching the laboratory, and they may or may not be recognized by the researcher [30]. Even if recognized, they may be difficult to adjust for in an analysis. Bias may also be introduced in the laboratory, producing results which may be related to artefacts of sample processing, but not due to disease specific pathology. Therefore, inequality of biospecimens remains to be the biggest flaw in the biomarker discovery. That can introduce bias early on in the studies, and 60–70% of all pre-analytical errors are due to the mishandling of samples during collection and processing [35]. Invalid proteomic and HER2 analyses [36] data from a clinical assay due to not adequately controlling the pre-analytical variables, is an example of a harmful outcome of using bio-specimens of poor or unknown quality.

The sample collection is a team effort, which typically involves patients, clinicians, researchers and biobank personnel. For a sample to truly represent a patient’s tumor biology, and to make a valuable contribution to biomarker discovery, it needs to be properly collected and stored. This crucially requires the coordinated work and communication between all the essential members in the team, who follow explicit, best practice guidelines in every step in the biospecimen’s journey, from the patient agreeing to donate it, to the specimen reaching the laboratory and going through the analysis. Therefore, it is important to bear in mind how a simple alteration, for example, tissue manipulation either before or after collection, can affect the end-result. Such aberrations could be erroneously reported as the changes in the expression levels of different genes and downstream targets, specific to a pathology. Therefore, it is an important responsibility of cancer researchers to remove such aberrations and to collect samples in optimal conditions. 

### 5.2. Biobanking Standards

Controlling pre-analytical variability is challenging and complex. However, as the quality of data obtained from the sample is directly dependent upon the pre-analytic factors, it is important to consider using the most appropriate samples and the most robust biomolecule analytical method to obtain useful data. In this respect, biobanks have an important role to play in adhering to stringent and explicit standards when handling samples. In general, most biobanks have their own standardized way of sample handling and local standard operating procedures (SOPs) and protocols. Unfortunately, between biobanks, there are wide variations in biobanking practices, such as the type of samples collected, sample quality, demographic data collected, ethical approval process, available patient consent, processing techniques and storage workflows. This can create challenges for the researchers to obtain suitable and comparable samples for collaborative research projects. 

The quality control of bio-banked samples can be regulated by (1) multidisciplinary scientific teams agreeing on the SOPs to adhere to at each stage of the biospecimen accruement, (2) standardizing these and communicating these with other scientific teams (3) by conducting specific-relevant research to identify new ways that will predict bio-sample integrity and quality [37,38,39].

### 5.3. The Factors/Issues Affecting Analytical Results

The results obtained from a biosample may be vulnerable to the quality of the biosample, in the context of the class of molecule analyzed, type of analytical method, the specificity, sensitivity and robustness of the method of analysis and the researcher controlling for the pre-analytical variables. Therefore, the researchers need to be fully aware of these issues pertinent to their samples and the employed methodology. Pre analytical variables such as biospecimen handling (e.g. snap freezing a sample immediately after collection as opposed to being transported in room temperature for several hours before freezing) may have an obvious effect on the integrity of the biospecimen and consequently on the downstream analyses [40]. Similarly, freeze thawing of samples after their acquisition in the laboratory can also affect the results, and should be considered by the researchers. Employing multiple techniques to confirm the data obtained from a biospecimen, examining multiple specimens from the same patient, and using a large number of samples from different sources to verify and to test the reproducibility of the results, are ways to reduce these pre/post-analytical biases in studies.

Genomics studies and transcriptomic analysis (e.g., microarrays/polymerase chain reaction (PCR)) depend on the sample stability and preservation of RNA integrity, hence even small temperature changes in collection, processing and storage can affect the scientific results. The SOPs used in different biobanks vary, depending on local resources, thus the quality of the samples can also differ widely. Implementation of quality management systems in biobanks and standardizing the best practice can lead to minimize the influence of these variations. Researchers can then obtain comparable samples for their research and conduct collaborative studies whilst facilitating the external validation of the promising results generated in smaller studies. 

Presently, analysis of big-data at high throughput speed is revealed in the scientific world, and thus importantly, considering biospecimens, we should strive to focus on quality, rather than the quantity to prevent wastage of time and resources. SOPs should be developed with input from all stakeholders and implemented in biobanks to minimize the variability while improving quality. This will encourage consideration of all possible but rectifiable aspects in sample handling. 

Making explicit records of the pre-analytic variables the specimens are subjected to is called the pre-analytical characterization. They should be part of the documentation held in biobanks, as they allow accurate grouping of similar samples during analysis.

## 6. Role of Harmonization of Biobanking and Existing Initiatives 

As previously mentioned, many agencies have recognized the importance of the harmonization of the biobanking of human biological samples (Table 1). The welcome trust case control consortium and The Cancer Genome Atlas (TCGA) project have recognized issues with inconsistent sample quality. Different data from different sources [41] recommend consistency with biospecimens quality [42]. However, studies still report difficulties in obtaining sufficient high-quality bio-samples of diseased and control biological materials to come to definite conclusions [43,44]. 

The European Prospective Investigation into Cancer study coordinated from the International Agency for Research on Cancer, and the Telethon Network of Genetic Biobanks in Italy, have ventured into standardizing the SOPs, their consent, transfer policies and procedures [45,46]. The European strategy forum on research infrastructures recognized that major synergy, gain of statistical power and economy of scale is by interlinking, standardizing, harmonizing or just cross referencing with a large variety of well qualified, existing, up-to-date national resources [47]. This foresaw the development of the Biobanking and Biomolecular resources Research Infrastructure [48]. International biobanking platforms like ‘The Marble Arch International Working Group on Biobanking for Biomedical Research’ and the ‘International society for Biological and Environmental Repositories’ have also been working on standardization of biobanking at global level [49,50,51]. More than a decade ago, NCI launched an investigation to understand the state of resources and the quality of biospecimens used in cancer research, developing a detailed NCI-best practice guidance for biobanks [52]. This has established guiding principles for practice, promoting biospecimen and data quality maintenance, and also details the ethical and legal considerations. Although their adaptation is voluntary, they support the optimization of the resources available for cancer research on a global level. 

Adapting and applying the current established best practice documents from some national institutes such as the ‘National Institute of Health/NCI’s Biorepositories and Biospecimen Research Branch Best Practices for Biospecimen Resources’, the ‘International Society of Biological and Environmental Repositories Best Practices for Repositories: Collection, Retrieval, and Distribution of Biological Materials for Research’ and the ‘World Health Organization International Agency for Research on Cancer Common Minimum Standards and Protocols for Biological Resource Centers Dedicated to Cancer Research’ will assist to improve the harmonization process for biobanks, and by raising the overall awareness and quality of research involving bio specimens. 

**Table 1 cancers-11-01734-t001:** List of International and National Efforts of Harmonization of biobanking.

Year	Project	Role	References
2003	Public Population Project in Genomics (P3G)	Not for profit international consortium, promoted collaboration between researchers in genomics.	[53]
2005	Wellcome Trust Case Control Consortium	UK wide consortium, explored utility, design and analyses of Genome wide association studies.	[54]
2005	International Society for Biological and Environmental Repositories (ISBER)	Global forum which addressed the harmonisation of scientific, technical, legal, ethical issues of repositories.	[51]
2006	The Cancer Genome Atlas Project	Cancer genomics program, a joint project between National cancer institute and National human genome research institute.	[55]
2006	European Human Frozen Tumor Tissue Bank TUBAFROST	Virtual European human frozen tumor tissue bank, has access to high quality tissue collections, which are made available for the researchers.	[56]
2006	International Agency of Research on Cancer (IARC)	International project funded by WHO, international collaboration on cancer research for cancer prevention.	[57]
2006	First-Generation Guidelines for NCI-Supported Biorepositories	National Cancer Institute (NCI) drafted guidelines to standardize and enhance the quality of research material collected by the repositories.	[58]
2007	Biobanking and Biomolecular Resources Infrastructure (BBMRI)	European network, with biobanking focus on human biosamples.	[33]
2014	World Endometriosis Research Foundation Endometriosis Phenome and Biobanking Harmonization Project (WERF EPhect)	International working group, which achieved global consensus in standardizing the data collection tools and protocols in endometriosis research.	[59]

The huge efforts that have already been made as described above, have ensued many individual biobanks to be well-organized and to be accessible bio-sample repositories. However, this is not a uniform process. The prevailing bank-specific variations are still too large to source samples from all biobanks to a single study and to generate robust results. Hence, further harmonization is a necessity to utilize the available resources to their maximum potential.

## 7. Need for Specific Tools for Collection of Accurate Data 

Biomarker-discovery studies have a wide variation and conflicting results, and these may be due to the lack of some essential data. For example, surgical phenotypic data, details of patient symptoms, together with other relevant information regarding sample handling that are specific to a particular condition/pathology, can influence the results. There is inconsistency in the type of data collected and the protocols used; hence, prior biobanked samples are not easy to be used in new, large, international, collaborative projects. This leads to the regular publication of a large number of studies with insufficient power that are simply ineffectual in making useful conclusions [59]. This huge waste in time and resources can be avoided with a harmonized biobanking practice that facilitates the easy organization of highly sought after, large scale, international, collaborative studies. Detailed surgical, clinical and epidemiological data pertinent to a specific cancer type can then be collected to accompany the biosamples from cancer patients, and thus will support many scientifically valid enquiries, producing a maximum return from the resources employed in sample collection. For standardization, it is important to have SOPs for the collection, processing and storage of the particular biospecimens and their accompanying clinical, surgical and other relevant data. 

The surgical team is best placed to collect important clinical/surgical information, for example, intraoperative findings relevant to the clinical-staging of the cancer and complications. They will help to enlighten researchers both in basic science and in clinical research. This can be a time consuming process imposed on the surgical team, to collect and record the data. However, the engagement of the surgical team validates the clinical details, and they may also contribute their knowledge and understanding of the disease to link scientific discoveries with clinical outcomes. Information about clinical symptoms, previous relevant medical history of the patient, up to date comorbidities, medications, etc. can be directly acquired from the patients who are the most accurate source of information in those aspects. This can be done by means of a self-completed questionnaire. As previously mentioned, it is important to recognize and minimize the variability by standardizing the collection, processing and storage of biological samples. Formulation of SOPs in advance, which are diligently adhered to by the biobank personnel is thus warranted. 

## 8. Methods of Harmonization 

Previous research has used several consensus generation methods, which mainly aim to achieve agreement of opinion on a particular topic, especially where published literature is inadequate. Consequently these methods have been generally used for problem solving or idea generation [60]. Three main consensus generation methods commonly adapted are: (i) The nominal group technique, (ii) consensus development conference and (iii) Delphi process. These methodologies are generally helpful in overcoming the disadvantages relevant to other less favored methods, such as committee meetings that can be dominated by one person or a group, usually with stakes or perception bias. In contrast, the focus of the more acceptable and thus favored consensus generation methods, is to assess the extent of agreement and to resolve disagreement. Therefore, the final result of these main methods is the inclusive and comprehensive agreement. 

### 8.1. Nominal Group Technique 

The nominal group technique is a structured face-to-face meeting where the panelists rate, discuss and rerate a number of questions. This has been mainly used in assessing the appropriateness of clinical interventions, education, training, and practice developments in the healthcare setting [61]. 

### 8.2. Consensus Development Conference 

Consensus development conferences have been used for safety, effectiveness and appropriateness of medical care and technology. These are run informally in terms of criteria for generating consensus. The definition of consensus in these conferences is unanimous agreement with the consensus statement. The processes used can affect the value, validity and hence its reproducibility [62]. As in any consensus generation methodology, investment of time is necessary, but this is particularly true with conferences, as conferences need to be organized, and participants have to attend all of the meetings to ensure its reliability and validity in reaching a consensus. Hence, this method has the added complexity of being more expensive and time consuming. Debates and disagreements during the consensus generation conferences may deviate the attention and focus of the entire group [62]. Some of the well-known examples of the use of this method are those conducted by the National institute of Health (NIH) [63] and World endometriosis research foundation endometriosis phenome and biobanking harmonization project [59]. 

### 8.3. Delphi Process 

The Delphi process is also a structured process, but here, the interaction amongst the panel members is through questionnaires, hence preserving anonymity. In this Delphi process, relevant individuals are invited to provide opinions, and they are also invited to participate in responding to different rounds of questionnaires. During each round, opinions are grouped together, and the questionnaire is redrafted until consensus is achieved on all topics included in the questionnaire. Although this appears to be a vigorous and sensible approach, there have also been various criticisms, such as the lack of evidence on the reliability of the Delphi process and its validity. Poor response rates may impede the process, and this is also another limitation of this method, and currently there are no guidelines on the number of consultation rounds that should be used as a standard, hence process can be variable [64]. 

Ironically, there is no general consensus or clear evidence as to which consensus generating method out of the above, is the best. Therefore, usually in a particular study, it is sufficient to clearly justify the reasons for choosing the specific method, and to present the findings and their relevance in the context of the method [61]. 

## 9. Key Stakeholders in Cancer Research 

Participation of all key stakeholders is paramount in the consensus generation exercise since they are the end users, which increases the acceptability through a sense of ownership and engagement. The benefit of involving patients is increasingly recognized [65] and includes procuring more accurate and reliable personal, past medical and past surgical data for research. Examples of information which can be obtained from patients include current weight, lifestyle data and family history. Although patients are likely to recall personal information on past medical, surgical and family histories more accurately, self-completed patient questionnaires used in this regard should be adequately prepared, with extensive patient/public involvement. This is because the accuracy of the information gathered is dependent upon the acceptability, user friendliness and clarity of the questions. A stringent methodology in developing the patient questionnaires and testing its reproducibility, suitability and acceptability to patients from different social, ethnic and cultural backgrounds, is therefore essential. 

The healthcare professional involvement in basic science research is vital, since they are instrumental in translating scientific discoveries into clinical practice. They can convey the relevance of the research to patients well, and thus recruit participants appropriately for the studies. Clinicians with adequate knowledge/experience can ensure documenting and verifying accurate clinical-surgical information, such as details about a particular operation, surgical findings, cancer staging and patient follow up data relevant to cancer research [66]. Clinicians partake in the current standard clinical management pathway, and they can thus bridge the gap between basic and applied research [67].

Pathologists make the diagnostic confirmation of cancer, and they are key members of the biobanking team who procure the surplus clinical diagnostic material as biospecimens for research to be stored in biobanks. Pathologists thus are quality controllers of each sample, and contribute to developing robust SOPs for biospecimen collection, processing and storage [68]. 

Researchers and biobank personnel carry out fundamental work in specimen processing, storage and data management. Detailed information on the time of processing a sample, whether SOPs were accurately followed, if there were any deviations, and reasons for deviation from the protocol, are all valuable in harmonizing biospecimens, since that would allow comparison between groups and individuals. These are recorded by the biobank personnel for the samples collected, and they reduce bias and enable large scale collaborative studies [69].

## 10. Developing Tools for Harmonization of Biobanking Standards in Endometrial Cancer Research—HASTEN Study 

Suitably collected patient material stored to high standards in Biobanks allows the study of multiple aspects of a single EC tumor, using novel technological platforms in genomics, proteomics, epigenomics and metabolomics, thus to simultaneously generate a large amount of information. Such an all-encompassing approach is expected to considerably reduce the time taken for new basic scientific discoveries to reach patients in the form of new treatments, as well as allowing the samples donated by patients to be fully utilized. 

As described above, there are many generic biobanking standards and initiatives in place already. Although they are an important start, many parameters and variables of interest, including the choice of biospecimens and clinical data, are cancer-type specific. Thus, universal biobanking standards are not necessarily applicable to every cancer-type and should be adapted to each specific disease. The importance of a cancer-specific harmonization of biobanking standards is highlighted by the TCGA [70], which now contains over 532 EC samples with RNA sequencing, copy number variation, proteomic, mutation and microarray data. However, the extremely limited clinical data accompanying most of these samples and datasets severely affects the ability of researchers to draw clinically applicable information. 

Therefore, EC specific standardization of the collection of biospecimens with distinctive and relevant accompanying clinical data sets, was a fundamental unmet need in improving future EC research. This, we believe, will facilitate future large-scale internationally collaborative research into EC, which could lead to improved biomarker and target treatment method discovery. Similar harmonization projects have already been successfully implemented for other gynecological conditions such as endometriosis, with the World Endometriosis Research Foundation Endometriosis Phenome and Biobanking Harmonization Project (WERF EPHect), and the Ovarian Cancer Research Program (OVCARE) [71,72,73,74,75]. 

With this background, we initiated our study (Harmonization of biobAnking STandards in Endometrial caNcer research—HASTEN) in 2016, to achieve consensus amongst EC researchers. This was to standardize the collection, processing and storage of all relevant biospecimens, and the accompanying clinical data for EC research through a joint effort with all stakeholders of EC research. 

Harmonization of EC research required the inclusion of all the above-mentioned variables, which increases the risk of EC in a woman. We also considered the common variations in the sample collection process; for example, the samples could be obtained during the diagnostic process (as a pipelle/curettings, Figure 1) or during the therapeutic procedure (from a hysterectomy sample). Variations in the handling of the samples were also considered, for example, frozen samples of presumed EC may or may not contain actual cancer cells, but only the background/adjacent hyperplasia or normal endometrium (Figure 2); thus diligent confirmation of the actual phenotype of the cellular content included in the specimen by histological scrutiny is required.

After an initial, thorough literature search and a critical appraisal of the available current evidence, four tools were consequently developed. Local, regional and European consensus on these tools was obtained through a comprehensive consultation process. When the final versions of the harmonization tools were developed, and final consensus was generated by a modified Delphi system. The modified Delphi system included sending the tools to panel members representing all the stakeholders in EC research, which included patients, gynecological oncologists, researchers, pathologists and biobank staff. The tools went through several rounds of revision according to the comments received, until unanimous consensus was reached. The final tools developed are freely available for any researcher via open access publication and the European Network for Individualized Treatment in EC (ENITEC) website. They include an EC patient data collection tool, an EC surgical data collection tool, an EC biospecimen tool and a Standard operating procedure for the collection, processing and storage of tissue or fluid for EC research [76] (Figure 3).

### 10.1. Endometrial Cancer Patient Data Collection Tool (ECPD)

This user-friendly data collection tool captures important demographic variables that are relevant to EC research. These we believe can only be accurately recalled by the patients [76]. For example, the available literature suggests that >20 kg of adult weight gain to be independently associated with an increased risk of EC. However, this information is unlikely to be obtained easily from any other mean, but directly from the patient. Many other risk factors for EC, such as the age of presentation, the postmenopausal status, history of polycystic ovarian syndrome, nulliparity, early age of menarche, family history of hereditary lynch syndrome-related cancers, past history of lynch syndrome-related cancers, medical conditions such diabetes, previous use of tamoxifen and hormone replacement therapy and exercise habits, are similarly best recalled by the patient, and therefore are also included in the tool (Supplementary Document 1 in [76]). 

### 10.2. Endometrial Cancer Surgical Data Collection Tool (ECSD)

The surgical data tool includes salient demographic, histological and pre/postoperative features [76] relevant to EC. It also includes information about preoperative imaging details and preoperative investigations such as endometrial biopsy results. Immunohistochemical biomarkers can be used to distinguish ECs from ovarian, cervical or other malignancies, but importantly they may also serve as prognostic biomarkers that are associated with clinical outcome. This tool is organized into different sections containing; surgical data: To be completed at the time of sample collection; histopathology details: To be completed after final staging and treatment and; outcome data: To be documented during follow-up and finally at the end of follow-up period (Supplementary Documents 3 and 4 in [76]). 

### 10.3. Endometrial Cancer Biospecimen Tool (ECBS) 

Variations in the collection methods at the time of diagnosis or treatment, and biobanking variables such as processing and storage, may alter the molecular composition, expression and stability of biomarker profiles, and thus, consistency and strict adherence to SOPs is vital. Therefore, information regarding the times of processing, storage and any deviations from this SOP needs to be documented clearly by the biobanking personnel (Supplementary Document 2 in [76]). 

### 10.4. Standard Operating Procedure for Collection, Processing and Storage of Tissue or Fluid for Endometrial Cancer Research (SOP-ECBS)

The methods used for investigating different tissue and fluid biospecimens collected may involve the extraction of protein, RNA and DNA, using a variety of techniques, such as proteomics, genomics and metabolomics. A SOP amalgamating a number of separate, detailed methodological protocols (e.g., for centrifugation, filtration, addition of preservatives, as well as storage temperatures) is required, and therefore, was devised [76]. Studies examining specimens collected via non-invasive means, including saliva and urine, are of a particular interest in clinical research. Future work is expected to focus more on them, hence SOPs need to include all samples which could be collected for EC research and are considered in this tool (Supplementary Document 5 in [76]).

## 11. Conclusions

Incidences of EC, and EC associated surgical morbidity and mortality, are increasing at an alarming rate. The causative factors, such as obesity and longevity, with their associated co-morbidities, are only expected to increase in the future, adding further pressure on clinicians and researchers to find novel, personalized diagnostic, therapeutic, prognostic and preventative strategies. 

For EC research and personalized EC treatment to be benefitted from the advances in ‘Omics’ technology, robust and extensive repositories of patient derived biological samples with accompanying detailed surgical, clinical and epidemiological data, is essential. Thus, harmonization of biobanking standards is a vital step toward high quality standardized, large-scale international collaborative projects to generate data that is translated into personalized clinical practice. The HASTEN project devised EC-specific tools and SOP through a comprehensive consensus generation process for the first time, and these will provide the necessary guidance and means for all EC researchers to standardize biobanking EC-related biospecimens. This, we envisage, will be a significant step in improving the quality of EC research in general, and will in the future result in enhancing clinical care through personalized management for the benefit of many women suffering from EC. 

## Figures and Tables

**Figure 1 cancers-11-01734-f001:**
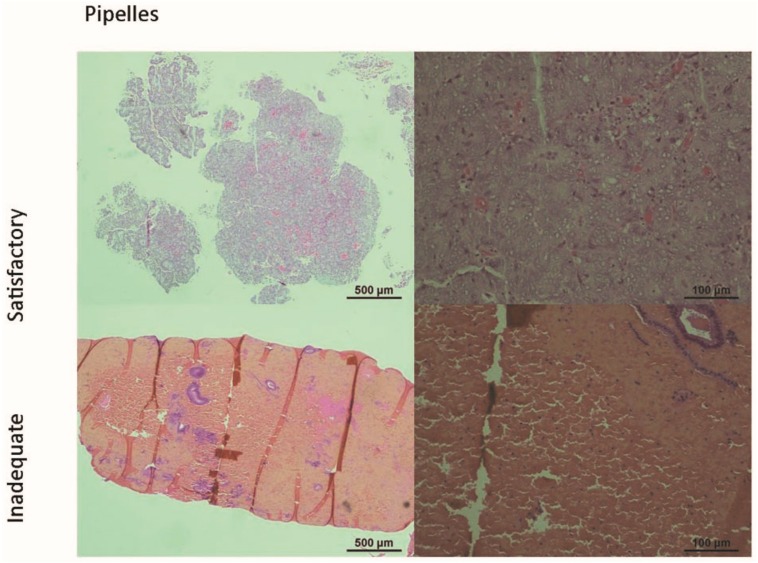
Micrographs of endometrial biopsies obtained during diagnostic procedures from patients with endometrial cancer. Pipelle (upper panel) and curettage (lower panel) samples may contain either satisfactory or inadequate amounts of cancer tissue as shown in this Haematoxylin- and Eosin-stained formalin fixed and paraffin-embedded tissue sections. This may be due to the skill of the clinician obtaining the sample, the endometrial thickness and the presence of mucus/blood, but they are inherent and unpredictable problems associated with these methods. Therefore, when a sample is collected by using these methods, and it is directly assigned for genomic and proteomic studies without confirming their cellular/tissue content, they may not produce credible data.

**Figure 2 cancers-11-01734-f002:**
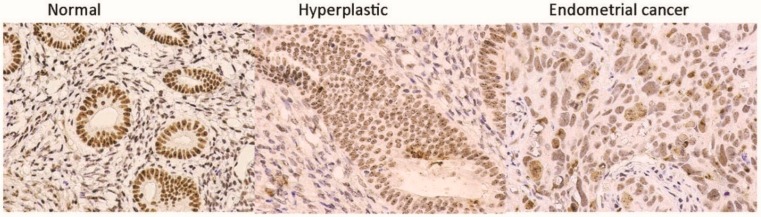
Micrographs of three separate endometrial samples obtained from the same patient, containing normal endometrial glands, hyperplastic glands and frank endometrial cancer tissue. The exact pathology included in the part of the sample studied with high throughput methods will directly influence the data generated. As shown here, the three separate parts of the endometrium biopsied from the same hysterectomy sample contained a histologically different pathological phenotype in the epithelial cells.

**Figure 3 cancers-11-01734-f003:**
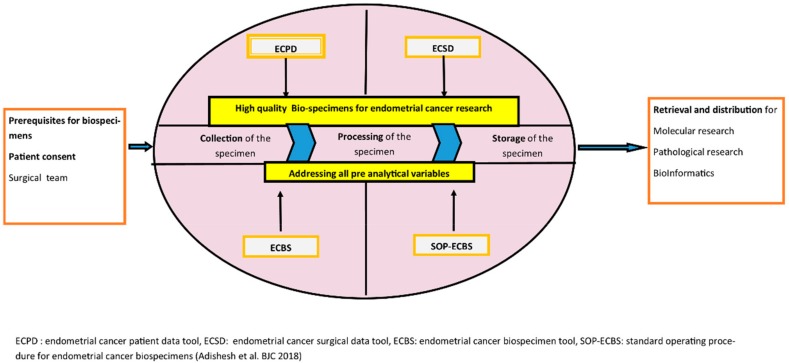
Journey of a biospecimen. Schematic representation of the utility of the tools developed with the HASTEN study (Harmonization of biobAnking STandards in Endometrial caNcer research). ECPD: Endometrial cancer patient data tool, ECSD: Endometrial cancer surgical data tool, ECBS: Endometrial cancer biospecimen tool, SOP-ECBS: Standard operating procedure for endometrial cancer biospecimens (Adishesh et al. BJC 2018).

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
