# Peer review of "Enriching Personalized Endometrial Cancer Research with the Harmonization of Biobanking Standards"

_cancers, 2019, doi:10.3390/cancers11111734_

Round 1
Reviewer 1 Report
Nice paper
I only have some minor comments:
line 34-36: explain possible reasons why incidence is rising
line 47-50: explain effects ER alpha and ER beta in this context
The bottom line is rigourous documentation and standardisation of all phases in the process: sampling methods, the lab processing, analytical and aquisition of clinical data
Author Response
Reviewer 1
Nice paper. I only have some minor comments:
line 34-36: explain possible reasons why incidence is rising
We thank the reviewer for the comment and have now moved the reasons for increased incidence of EC to be placed next to this introductory paragraph, thus explanations are in close proximity to the statement. Furthermore, we revised the sentence to reference the reasons, page 1, line 35; “The rise in EC rates is a global phenomenon, as shown by European and North American studies due to reasons detailed later in this review.”
line 47-50: explain effects ER alpha and ER beta in this context
We have revised the section “Estrogen exerts its trophic/mitotic effects mainly via estrogen receptor alpha (ERα), whilst ligand-activated ERβ counteracts and regulates ERα action [5].
The bottom line is rigourous documentation and standardisation of all phases in the process: sampling methods, the lab processing, analytical and aquisition of clinical data
We agree and thank the reviewer for highlighting the importance of the message we try to convey in this review.
Reviewer 2 Report
This manuscript is a well documented review. The manuscript is suitable to publication in cancers.
Author Response
Reviewer 2
This manuscript is a well documented review. The manuscript is suitable to publication in cancers.
We agree and thank the reviewer for highlighting the importance of the message we try to convey in this review.
Reviewer 3 Report
The review entitled “Enriching personalized endometrial cancer research 2 with the harmonization of biobanking standards” by Adishesh and Hapangama has addressed the
Critical issues in biobanking in EC, which is timely important.
The review covers the different aspects of biobanking and is highly informative. However, it is a bit dis-organized. A better logic presentation of the current information will increase the readability of the review. In addition, the word usage/orders in sentences and the presentation style need to be improved. Some specific commends are listed below.
Move Section 9 “Reasons for the increase in the incidence of EC” to section 2, which is a logic flow form Introduction. In “introduction, the subtitle “Endometrial cancer” should be removed, since there is no additional subtitle in this section. Line 34: “… mortality rates from EC to have gone up by 21%...”. Remove “to” from the sentence. Lines 46-47: “…hormone responsive organs, there are further striking dissimilarities in the responses of endometrial cells to those hormones.” Add, “in human vs. other mammals” to indicate dissimilarities between what. Line 52: “…expansion of EC research.” Add “using human samples” to the end to emphasize the importance of human samples. Line 67: “However, patient derived ….” The “however may be changed to “In addition”. Lines 99-100” Therefore, the commonest way the samples are still processed and stored is by preservation in fixative …” Move “still” after “is” in this sentence. Lines 344-346, :Though it has anti estrogenic effects on breast tissue, moderate estrogenic effects are seen on endometrium, hence in standard doses it causes endometrial proliferation leading to hyperplasia, polyp formation and invasive cancer Line 381-383: “As described above, there are many generic biobanking standards and initiatives in place 381 already. Although this is an important start …” Change “this is” to “they are”. Lines 398-399: With this background, we initiated our study (Harmonization of biobAnking STandards in Endometrial caNcer research - HASTEN). “ Is this an ongoing project? If yes, use “have initiated” and give the time that has been initiated. Section 4.3. The analytic The Title needs to be changed to something like “the factors/issues affecting analytical results”. Capitalization of title/subtitles are inconsistently used. Fig 1 needs to be better labelled for the panels and explain why some samples are inadequate. The reference style needs to be more consistent and needs to meet the journals requirements.
A more careful check in the entire manuscript is needed, which should not be limited to the specific issues listed above.
Author Response
Reviewer 3
The review entitled “Enriching personalized endometrial cancer research 2 with the harmonization of biobanking standards” by Adishesh and Hapangama has addressed the Critical issues in biobanking in EC, which is timely important.
We thank the reviewer for highlighting the importance of the message we try to convey in this review.
The review covers the different aspects of biobanking and is highly informative. However, it is a bit dis-organized. A better logic presentation of the current information will increase the readability of the review.
We thank the reviewer, considered the comments and re-organised the presentation of the review, checked it for errors and edited accordingly.
In addition, the word usage/orders in sentences and the presentation style need to be improved. Some specific commends are listed below.
We thank the reviewer, considered the comments and revised the sentences extensively and re-organised the presentation style of the review.
Move Section 9 “Reasons for the increase in the incidence of EC” to section 2, which is a logic flow form Introduction.
We agree and the manuscript is now revised accordingly.
In “introduction, the subtitle “Endometrial cancer” should be removed, since there is no additional subtitle in this section.
We agree and the manuscript is now revised accordingly.
Line 34: “… mortality rates from EC to have gone up by 21%...”. Remove “to” from the sentence.
We agree and the manuscript is revised accordingly.
Lines 46-47: “…hormone responsive organs, there are further striking dissimilarities in the responses of endometrial cells to those hormones.” Add, “in human vs. other mammals” to indicate dissimilarities between what.
We agree and the manuscript is revised accordingly.
Line 52: “…expansion of EC research.” Add “using human samples” to the end to emphasize the importance of human samples.
We agree and the manuscript revised accordingly.
Line 67: “However, patient derived ….” The “however may be changed to “In addition”.
We agree and the manuscript revised accordingly.
Lines 99-100” Therefore, the commonest way the samples are still processed and stored is by preservation in fixative …” Move “still” after “is” in this sentence.
We agree and the manuscript is revised accordingly.
Lines 344-346, :Though it has anti estrogenic effects on breast tissue, moderate estrogenic effects are seen on endometrium, hence in standard doses it causes endometrial proliferation leading to hyperplasia, polyp formation and invasive cancer
We agree and revised it to “On breast tissue, tamoxifen has anti-estrogenic effects while moderate estrogenic effects are seen on the endometrium, therefore, in standard doses it causes endometrial proliferation leading to hyperplasia, polyp formation and invasive cancer”
Line 381-383: “As described above, there are many generic biobanking standards and initiatives in place 381 already. Although this is an important start …” Change “this is” to “they are”.
We agree and the manuscript is revised accordingly.
Lines 398-399: With this background, we initiated our study (Harmonization of biobAnking STandards in Endometrial caNcer research - HASTEN). “ Is this an ongoing project? If yes, use “have initiated” and give the time that has been initiated.
We agree and the manuscript is now revised accordingly “With this background, we initiated our study (Harmonization of biobAnking STandards in Endometrial caNcer research - HASTEN) in 2016 to achieve…”
Section 4.3. The analytic The Title needs to be changed to something like “the factors/issues affecting analytical results”.
We agree and the manuscript is revised accordingly. The new title reads “The factors/issues affecting analytical results”
Capitalization of title/subtitles are inconsistently used.
We apologise for this error and checked and revised the titles and subtitles ensuring consistency
Fig 1 needs to be better labelled for the panels and explain why some samples are inadequate.
We agree and now Fig 1 legend is revised to “This may be due to the skill of the clinician obtaining the sample, endometrial thickness, and presence of mucus/blood but they are inherent and unpredictable problems associated with these methods. Therefore, when a sample collected using these methods is directly assigned for genomic and proteomic studies without confirming their cellular/ tissue content, they may not produce credible data.”
The reference style needs to be more consistent and needs to meet the journals requirements.
We agree and the manuscript is revised accordingly.
A more careful check in the entire manuscript is needed, which should not be limited to the specific issues listed above.
We agree and the manuscript has been thoroughly checked and is revised accordingly.
Round 2
Reviewer 3 Report
The revision has addressed my concerns and it is now acceptable for publication in my opinion.